# The Lived Experience Of Participants in an African RandomiseD trial (LEOPARD): protocol for an in-depth qualitative study within a multisite randomised controlled trial for HIV-associated cryptococcal meningitis

David S Lawrence [1,2] Katlego Tsholo,[1] Agnes Ssali,[3] Zivai Mupambireyi,[4] Graeme Hoddinott [5] Deborah Nyirenda,[6] David B Meya,[7] Chiratidzo Ndhlovu,[8] Thomas S Harrison,[9] Joseph N Jarvis,[1,2] Janet Seeley [2,3]

JNJ and JS contributed equally.

**Correspondence to**
Dr David S Lawrence;
david.s.lawrence@lshtm.ac.uk

## ABSTRACT

**Introduction** Individuals recruited into clinical trials for life-threatening illnesses are particularly vulnerable. This is especially true in low-income settings. The decision to enrol may be influenced by existing inequalities, poor healthcare infrastructure and fear of death. Where patients are confused or unconscious the responsibility for this decision falls to relatives. This qualitative study is nested in the ongoing AMBIsome Therapy Induction OptimisatioN (AMBITION) Trial. AMBITION is recruiting participants from five countries in sub-Saharan Africa and is trialling a novel treatment approach for HIV-associated cryptococcal meningitis, an infection known to affect brain function. We aim to learn from the experiences of participants, relatives and researchers involved in AMBITION.

**Methods and analysis** We will collect data through in-depth interviews with trial participants and the next of kin of participants who were confused at enrolment and therefore provided surrogate consent. Data will be collected in Gaborone, Botswana; Kampala, Uganda and Harare, Zimbabwe. Interviews will follow a narrative approach including participatory drawing of participation timelines. This will be supplemented by direct observation of the research process at each of the three recruiting hospitals. Interviews will also take place with researchers from the African and European institutions that form the partnership through which the trial is administered. Interviews will be transcribed verbatim, translated (if necessary) and organised thematically for narrative analysis.

**Ethics and dissemination** This study has been approved by the Health Research Development Committee, Gaborone (Reference: HPDME:13/18/1); Makerere School of Health Sciences Institutional Review Board, Kampala (Reference: 2019–061); University of Zimbabwe Joint Research Ethics Committee, Harare (Reference: 219/19), and the London School of Hygiene and Tropical Medicine (Reference: 17957). Study findings will be shared with research participants from the sites, key stakeholders at each research institution and ministries of health to

## Strengths and limitations of this study

► There has been no previous qualitative study conducted in a low-income setting which has aimed to explore the experience of individuals who enrol into a clinical trial for the management of a life-threatening illness.

► We plan to collect data from trial participants, their next of kin and researchers working on a multisite clinical trial and by doing this we can elicit a broad range of perspectives and experiences that can inform the improvement of this and similar trials in the future.

► By adopting a multisite approach, we can compare and contrast experiences across different settings to understand which are shared and which are unique to a particular context.

► The study team are from multiple social and behavioural science disciplines meaning that interpretation of the data will be informed by a range of social theoretical perspectives.

► This study is taking place in a single clinical trial and will collect data from individuals in Botswana, Uganda and Zimbabwe only which means that the results may not be broadly generalisable.

help inform the development and implementation of future trials. The findings of this study will be published in journals and presented at academic meetings.

**Trial registration** Registered at www.clinicaltrials.gov: NCT04296292.

## INTRODUCTION

Since the start of the HIV epidemic our knowledge and understanding of the epidemiology and management of HIV and its numerous complications has exponentially increased. This knowledge has been produced through

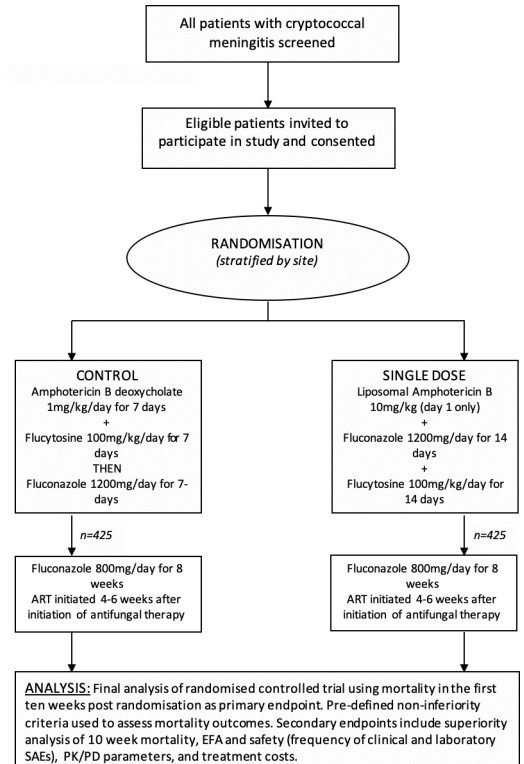

**Figure 1** AMBITION Trial schema. ART: antiretroviral therapy, EFA: Early Fungicidal Activity, SAEs: Serious Adverse Events, PK/PD: Pharmacokinetics/Pharmacodynamics

the conduct of clinical research which would not be possible without the willing consent of participants.[1 2] Although antiretroviral therapy (ART) programmes have expanded dramatically and AIDS-related deaths have reduced, there were still an estimated 940 000 people who died from AIDS in 2017.[3] In individuals with advanced HIV disease the search for superior treatment options for fatal opportunistic infections continues.

### The AMBITION Trial

The AMBIsome Therapy Induction OptimisatioN (AMBITION) Trial is a phase-III multicentred randomised controlled trial recruiting patients with HIV-associated cryptococcal meningitis (CM)[4] (figure 1). CM is a fungal infection of the brain that occurs most frequently in severely immunocompromised individuals with a CD4 count of less than 100 cells/uL.[4] There are approximately 223 000 incident cases of CM globally, with 73% of these occurring in sub-Saharan Africa. Annual global deaths are estimated at 180 000 and CM is responsible for roughly 15% of all AIDS-related deaths.[5] The nature of the infection means that roughly 40% of patients present with confusion[6] and some with a significantly reduced level of consciousness.

AMBITION is testing a new treatment for CM, a single, high dose of a less toxic, liposomal form of amphotericin, and is recruiting 850 participants from eight hospitals across five African countries: Botswana, Malawi, South Africa, Uganda and Zimbabwe. The decision-making capacity of potential participants is assessed by the clinical

team who determine if the individual is able to understand the information around the trial, retain that information, weigh up the information to make a decision and communicate that decision. Patients consent for themselves if deemed to have decision-making capacity and if they do not, for example, if they are confused or comatose, then a surrogate will do so on their behalf. Participants are followed up daily during their initial inpatient admission (roughly 2 weeks) and then fortnightly as an outpatient until they complete the study at 10 weeks. Participants have their medical expenses paid for and receive transport reimbursements to attend outpatient appointments. AMBITION is funded by the European and Developing Countries Clinical Trials Partnership which brings together researchers from institutions in low and middle income countries (LMICs) and Europe.

The AMBITION Trial creates a rich environment for an in-depth qualitative study for a number of reasons.

### Why participants are motivated to participate in trials

In routine care, mortality with the best standard of care treatment for CM is roughly 50% at 10 weeks.[7] In recent CM trials using the same regimen mortality is roughly 40%.[6 8] It has been observed that even when using the same drugs as in routine care, trial participants often do better.[7] The reasons for this include having a dedicated research team with more time for patients, better management of drug-induced toxicities and aggressive management of raised intracranial pressure, a common and potentially fatal complication of CM, and, inevitably, some selection of trial participants. Further widening outcomes between routine and trial settings in CM is the fact that the most effective drugs may be unavailable, or only sporadically available, in routine care. Clinical trials are however designed to answer a research question, the findings of which it is hoped will later be of benefit to a larger population. Some individuals may benefit by participating but it is not designed so that everyone will.[9] Despite this it is not uncommon for research participants to expect a personal therapeutic benefit from the treatment they receive, including in placebo-controlled trials.[10 11] Other commonly identified motivators are material benefits including free healthcare and transport reimbursements,[12–14] and altruism is also a factor.[15] In AMBITION it is fair to expect that all participants will benefit, compared with routine care. What is not understood is how this impacts both patients and researchers when it comes to motivating to enrol in the trial. Their motivation may be rooted in the economic inequality that exists between the patient and the research institution and which permeates the concept of voluntary participation. Voluntariness is understood as an autonomous choice without material entanglements and the principle of autonomy is often held above others when it comes to consenting for a clinical trial.[16] Research participants who lack agency are therefore subject to 'structural coercion' whereby their social and economic situation drives them into research participation as a means of navigating their

illness and because they may not have any other options to get the care they need or desire.[17] This is polarised when the chance of death is high, such as in CM.

## Whom to consent when the patient cannot

In the context of life-threatening illness there are questions about when to obtain consent and who to obtain it from. One option is to commence trial procedures and defer consent until the patient is stable, which was acceptable to 70% of parents in a UK-based emergency paediatric study who felt the process was too much to handle in a stressful situation.[18] These findings are consistent with other studies from the UK.[19–21] The Declaration of Helsinki states that it is acceptable to recruit someone without capacity in best interests[22] and it has been argued that delaying treatment while waiting for consent risks losing out on the potential health benefit of that specific emergency treatment and underappreciating the impact of emergency treatment due to systematically delayed initiation.[23] An alternative is therefore to waiver informed consent completely, as was the approach in a postpartum haemorrhage trial in the UK which found that the perceptions of those who gave consent, had a surrogate, or waived consent were not dissimilar.[11]

Regarding who provides the consent, it is typical for surrogates to consent on behalf of an unwell patient who is confused or comatose. Within CM studies, roughly 40% of participants are confused and if they regain capacity they reconsent for themselves. Research in high-income countries (HICs) has identified that there is generally good concordance between surrogates and patients when it comes to agreeing to consent to both real life and hypothetical trials but that this is reduced in high-risk trials.[24 25] In LMICs multiple actors are often involved in the consent process with partners, parents, older family members and community leaders being consulted,[10 26] particularly in the case of severe illness.[13] This extends the process of gaining consent and can delay recruitment and treatment. According to a systematic review of 21 studies in Africa, only 47% of participants undergoing informed consent understood trial procedures such as randomisation and placebo and only 30% were aware they may not experience a therapeutic benefit of participation.[27] Another review found that understanding is significantly diminished among those who are critically ill.[28] To date there have been no in-depth qualitative studies in LMICs exploring the process of consent from the perspective of an acutely unwell adult or their consenting next of kin.

## Participant and next of kin experience

We use the broad term of participant experience to encompass the way that an individual navigates through the scheduled events of a clinical trial as detailed in the protocol. Time is a prominent factor throughout this process. An illness occurs at a specific time in someone's life and the entire trial experience is time bound and shaped by the protocolised schedule of events. A large portion of the ethnographic work exploring participant experience of research in LMICs has elicited data concerning rumours, most commonly blood stealing, which are often dismissed by researchers as expressions of ignorance but are interpreted by social scientists as forms of popular resistance.[29–31] Most ethnographic exploration of rumours has been situated in trials of healthy individuals in trials and less commonly in acute, life-threatening illness. Lumbar puncture, the procedure used to diagnose and treat CM is known to be associated with rumours of causing death.[32] This has not been extensively studied using ethnographic methods but lumbar puncture refusal is common and can be fatal.

In the USA there has been an increasing call to assess clinical trial participant 'patient satisfaction' through the use of surveys or interviews which aim to hear the participant's voice and respond by making local improvement to the trial.[33] In LMICs this approach is less common but the concept of 'good participatory practice' has been developed by the WHO over the years[34] and this involves elements related to the participant experience.[35] No ethnographic work has explored these in the context of acute illness research in sub-Saharan Africa. Research within healthy volunteer studies has found that where poor outcomes such as severe disability or death occur, this has led to the apportioning of blame or the generation of rumours about research studies and institutions.[29 36] An exploration within AMBITION, where poor outcomes are not uncommon, could provide an opportunity to inform and potentially improve the conduct of this trial and others in the future.

## Researcher experience

Paul Farmer (2002) wrote that 'researcher and subject are living in different worlds'[37] and it is commonly perceived that there is a mismatch between researcher and participant understanding of the research process.[29] Large, randomised controlled trials like AMBITION employ a large number of individuals from different countries.[38] Clinical researchers interact with individuals and their next of kin throughout the trial timeline[14 26 39 40] and are well placed to comment on the research process, regulatory approvals and implementation of a trial. These individuals can therefore provide a practical insight and suggestions for improvement.[41] As partners in the research process they can reflect on how clinical trials are conceptualised and designed in addition to the benefits and shortcomings of transnational partnerships and how we can optimise these relationships for the benefit of participants.[42] International researchers often have a broad range of experience working in clinical trials and can reflect on the evolution of clinical trials over time. As representatives of institutions which are partners (and often the lead) on grant applications, they often help to steer the clinical trial agenda in the region and are well placed to comment on how trials can be improved.

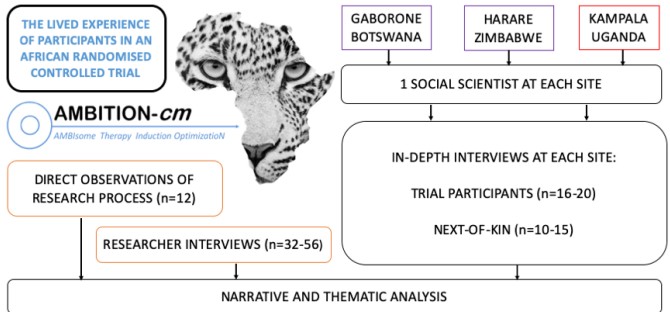

**Figure 2** The Lived Experience Of Participants in an African RandomiseD trial (LEOPARD) Study Schema.

## Aim and objectives

The aim of this study is to explore the experience of participants, their next of kin and researchers within the AMBITION Trial. By doing this we hope to learn how we can improve the trial experience within AMBITION and future trials for life-threatening illnesses.

Our specific objectives are:

From the perspectives of the participant, next of kin and researcher:

1. To build an understanding of the factors that enhance or diminish a clinical trial experience.
   From the perspective of the researcher:
2. To compare the individual researcher's experience of the conceptualisation, development, initiation and implementation of a multicentred clinical trial in Africa.

## METHODS AND ANALYSIS
### Study design

We propose an in-depth qualitative study entitled the Lived Experience Of Participants in an African RandomiseD trial (LEOPARD). We will adopt a combination of in-depth interviews (IDIs) and direct observations to explore the experience of participants, their next of kin and researchers within the AMBITION Trial (figure 2).

## Developing the methodology

The LEOPARD Study was conceived by DSL but the methodology was refined with the valuable input of social scientists from each of the six AMBITION Trial sites. Each social scientist has a particular interest in clinical trials. Over a series of one-on-one discussions and group calls the LEOPARD Study evolved. Having developed a consensus on a methodology it was necessary to determine the feasibility of collecting data from six sites. Recruitment into the LEOPARD Study will take place in Gaborone, Harare and Kampala. The reason for limiting data collection to three sites is to enable in-depth data collection and to avoid simply skimming the surface by diluting down data collection across multiple sites. These three sites represent diverse HIV epidemics, healthcare systems and political contexts which can be explored during data analysis.

## Conceptual framework

We will use narrative analysis to explore how the concept of time shapes the experience of a life-threatening illness and a clinical trial. Drawing on Nancy Munn's theory of temporalisation we will look at how time is experienced by different individuals and how the pressure of a life-threatening illness impacts the perception of time as well as the complex decision to enrol (or not) in a clinical trial. By understanding how time and pressure impact the meaning and understanding of events at a time of crisis, we hope to learn how clinical trials can be better tailored to the needs of individuals with life-threatening illnesses. Narrative analysis is more commonly adopted by studies exploring chronic health conditions but the exploration of time is well suited to narrative analysis and a clinical trial, which has a clearly defined temporal structure, provides a rich setting for story-telling.

## IDIs with AMBITION Trial participants

The purpose of the IDIs with AMBITION Trial participants is to collect personal accounts of their experience within the trial. Individuals who on entry into the AMBITION Trial were deemed to have decision-making capacity (ie, orientated) and those who were not (ie, disorientated), and therefore underwent surrogate consent, will be approached. All participants in the LEOPARD Study will need to have decision-making capacity to contribute to the IDI, meaning that those who lacked decision-making capacity at baseline will have clinically improved and regained that capacity. We will aim to recruit a maximum of 20 participants from each of the three sites, 60 in total, with a proposed gender balance of 50%–60% male and 40%–50% female which is in line with the epidemiology of CM at the sites. Consecutively eligible individuals will be approached to participate in two IDIs. One will take place at least 6 weeks into the 10-week AMBITION Trial and the other will take place at least 4 weeks after the trial. The reason for this is to allow reflection on the trial when one is both within and outside of it. Interviews will follow a broad interview schedule and the participant will be invited to draw a timeline of the events before, during and after the trial (online supplemental file 1). If individuals can only contribute to one IDI, for example due to worsening health or unavailability, then the data from the first IDI will be retained and analysed.

## IDIs with the next of kin of AMBITION Trial participants

The purpose of the IDIs with the next of kin of AMBITION Trial participants is to collect personal accounts from individuals who have cared for and made important decisions about someone with a life-threatening illness. We use the term next of kin as a broad umbrella term to include any individual who may be the legal representative, a caregiver or a surrogate of the participant. This individual will have provided consent for the participant to enrol into the AMBITION Trial even if they may not have been the legally defined next of kin. We will aim to recruit a maximum of 15 individuals from each site, 45 in total,

with no specification for gender. Consecutively eligible individuals will be approached to participate in a single IDI which will take place at least 6 weeks into the AMBITION Trial. At the time of the IDI it will not be necessary for the trial participant to have regained decision-making capacity and these IDIs do not need to be linked to those with participants, although it is anticipated that some will be. Interviews will again follow a broad interview schedule and the participant will be invited to draw a timeline of the events (online supplemental file 2).

## IDIs with AMBITION researchers

The purpose of the IDIs with AMBITION researchers is to understand their perspectives on how research is designed and implemented in Africa. Interviews will take place with researchers from each of the research institutions which form the AMBITION consortium. At African sites where trial participants are being recruited we will approach a range of individuals with different roles including senior and junior researchers, research doctors and nurses, laboratory scientists, pharmacists and study coordinators. In addition, individuals who are based at European institutions will be approached. We will aim for a maximum of 12 individuals from each of the three participating African sites and 4 from each of the five European sites. The maximum number of researcher interviews will be 56. Individuals will be conveniently sampled and interviewed on a single occasion, following a broad interview schedule (online supplemental file 3).

## Direct observations of AMBITION researchers

A period of 12 months will be spent conducting ethnographic fieldwork at the African sites. The objective of this work is to contextualise the data from IDIs within the broader research environment. As the primary focus is on improving the trial for participants, observations will be largely based in the clinical environment, with emphasis placed on observing clinical staff. A total of four researchers from each of the three African sites will be invited to participate in direct observations. It will be made clear that this is not a method designed to appraise an individual, but an opportunity to spend a defined period of time observing events that take place within the research process. Observations will be coupled with brief questions to those in close proximity to the activity under observation.

## Principles of recruitment

Eligible individuals will be approached to enrol in the study by a social scientist. In the case of AMBITION Trial participants and their next of kin, this will be done in the local language by an experienced social scientist at that site. In the case of AMBITION researchers this will be DSL who is part of the AMBITION Trial Management Group in his role as Lead Clinician for the trial. The researcher participant will be assured that they are free to decline participation and are not being interviewed or observed for the purposes of any appraisal or formal evaluation of their role within the team. The purpose of the researcher interviews and observations is to understand the research process and not to criticise individuals. A reflective approach to the research process will be adopted to iteratively refine the data collection methods and the communication skills of the social scientists.

Eligible individuals will be provided with a Participant Information Sheet and given the opportunity to ask questions. If they agree to participate, they will sign an Informed Consent Form and will be given the opportunity to withdraw their consent at any time, without giving a reason. Interviews will take place in a mutually acceptable location, be recorded with a digital voice recorder and notes will be taken during the interview. Observations will not be recorded and field notes will be made after the period of observation has finished.

It is anticipated that this study may identify aspects of the AMBITION Trial that need to be improved. In order to ensure this a formal reporting process will be followed. Each of the individual social science research assistants will report back to DSL. Any urgent issues that relate to trial conduct and Good Clinical Practice will be communicated through the use of direct communication and reflective summaries written on the day of data collection. In addition, weekly meetings will take place between the social scientists and DSL to discuss less urgent issues. These findings will be communicated either urgently to the Trial Management Group or at their weekly meetings, whichever is deemed appropriate. Additional advice may be sought from JS who is independent of the AMBITION Trial. Following this process the team will determine a course of action which may result, for example, in additional training of trial staff or modification of study procedures. This process is of vital importance to ensure that the findings of this study can improve the conduct of the ongoing AMBITION Trial. The confidentiality of the participant will be maintained throughout this process so as not to undermine trust in the study.

## Confidentiality

All study documents will be kept on the person of the researcher or in a secure, locked location at all times. All digital documents will be on a password-protected, encrypted computer, backed up regularly and only shared with the study team. Names of interviewees will not be used at any stage of the data collection process. Predetermined identification numbers will be used on data collection forms. Audio recordings will not start until the interviewee has given consent and will not record their name. Pseudonyms will be used throughout. The location of researcher participants will be anonymised because the small number of eligible participants means that stating their location could make it possible to identify them.

## Data analysis

Audio recordings will be transcribed verbatim into MS Word, translated into English in a separate second step if necessary, then exported to NVIVO V.11 for coding and

analysis. The first two IDIs from each group of participants will be analysed and discussed to enable iterative refinement of the data collection approach. Similarly, regular meetings will be used to review data, refine data collection tools and assess for data saturation. We will organise the data thematically and analyse it using narrative analysis at the country level by the social science team at each site. All data from AMBITION researchers will be analysed together using thematic analysis which will be performed in six phases: familiarisation with data, initial code generation, searching for themes, reviewing themes, defining and naming themes, and presenting final conclusions. These analyses will then be combined in a meta-synthesis of all data, irrespective of location or informant, to identify any areas of disconnect and, by comparing with country-specific analyses, to assess generalisability of findings.

## Patient and public involvement

This protocol has been reviewed by Community Advisory Board members, expert patients and HIV activists from across the African sites. These individuals and groups will continue to be consulted throughout the data collection process and during the dissemination of research findings.

## ETHICS AND DISSEMINATION

This study has been approved by the Human Resource Development Council, Gaborone (Reference HPDME:13/18/1); Makerere School of Health Sciences Institutional Review Board, Kampala (Reference: 2019–061); University of Zimbabwe Joint Research Ethics Committee, Harare (Reference: 219/19), and the London School of Hygiene and Tropical Medicine (Reference: 17957). Study findings will be shared with research participants from the African and European sites, key stakeholders at each research institution and ministries of health to help inform the development and implementation of future trials. The findings of this study will be published in journals and presented at academic meetings.

**Author affiliations**
[1]Botswana-Harvard AIDS Institute Partnership, Gaborone, Botswana
[2]Department of Clinical Research, London School of Hygiene and Tropical Medicine, London, UK
[3]Social Aspects of Health Programme, MRC/UVRI & LSHTM Uganda Research Unit, Entebbe, Uganda
[4]Centre for Sexual Health and HIV/AIDS Research, Harare, Zimbabwe
[5]Desmond Tutu TB Centre, Stellenbosch University Faculty of Medicine and Health Sciences, Cape Town, Western Cape, South Africa
[6]Malawi Liverpool Wellcome Trust Clinical Research Programme, Blantyre, Malawi
[7]Infectious Diseases Institute, Makerere University, Kampala, Uganda
[8]Department of Medicine, University of Zimbabwe College of Health Sciences, Harare, Zimbabwe
[9]Institute for Infection and Immunity, St George's University of London, London, UK

**Contributors** DSL wrote the initial manuscript and trial protocol and is the Chief Investigator for the study. KT, AS and ZM are social scientists based at each of the three African sites. GH is a social scientist in South Africa and contributed to the study design. DN is a social scientist in Malawi and contributed to the study design. DBM and CN are the AMBITION and LEOPARD principal investigators at the Kampala and Harare sites, respectively. TH and JNJ are co-Chief Investigators of the AMBITION Study. JS and JNJ jointly supervise DSL. All authors have been actively involved in the development of this study, the data collection tools and the analysis plan. All authors have reviewed and approved the final manuscript.

**Funding** This work is funded by the UK National Health Service (NHS) National Institute for Health Research (NIHR), using Official Development Assistant (ODA) funding through a Global Health Professorship to JNJ (Grant RP-2017–08-ST2-012). The views expressed are those of the authors and not necessarily those of the NHS, NIHR, the Department of Health and Social Care, or other funding entities.

**Competing interests** None declared.

**Patient consent for publication** Not required.

**Provenance and peer review** Not commissioned; externally peer reviewed.

**ORCID iDs**
David S Lawrence http://orcid.org/0000-0002-5439-4039
Graeme Hoddinott http://orcid.org/0000-0001-5915-8126
Janet Seeley http://orcid.org/0000-0002-0583-5272

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
