## [Reviewer comments · BMJ Open]

ARTICLE DETAILS

TITLE (PROVISIONAL)	The Lived Experience Of Participants in an African Randomised trial (LEOPARD): Protocol for an in-depth qualitative study within a multi-site randomised controlled trial for HIV-associated cryptococcal meningitis.
AUTHORS	Lawrence, David; Tsholo, Katlego; Ssali, Agnes; Mupambireyi, Zivai; Hoddinott, Graeme; Nyirenda, Deborah; Meya, David; Ndhlovu, Chiratidzo; Harrison, Thomas; Jarvis, Joseph; Seeley, Janet

VERSION 1 – REVIEW

REVIEWER	Laio Magno Universidade do Estado da Bahia, Brazil
REVIEW RETURNED	23-Jun-2020

GENERAL COMMENTS	It is a study with an excellent description of the methodological and very relevant procedures. I find it interesting to insert the objective of the manuscript at the end of the introduction. The authors justified the importance of the analysis “rumors” and “ethical implications”, but it does not bring these two topics to the objectives related to the study. I suggest adding. Are detailed interview scripts available? Can they be made available as an attachment to this article? Is there a script for direct observations? Can it also be made available as an attachment to this article? It is interesting to contextualize the three sites that will be included in the survey.
--

REVIEWER	Stuart Rennie University of North Carolina-Chapel Hill, USA
REVIEW RETURNED	09-Dec-2020

GENERAL COMMENTS	The qualitative study described is likely to produce valuable information about clinical trial participation among those involved in studies of life-threatening conditions in low- and middle-income settings, such as their trial sites in sub-Saharan Africa. I only have a few comments and questions. Major comments How will decisional (in)capacity, the key inclusion/exclusion criterion for patient-participants, be determined? For reasons that the protocol itself states, what will the qualitative researchers do if they find that the quality of informed consent among participants or next-of-kin for the AMBITION study is poor, either in terms of what participants understand or in terms of their
---

	ability (given their circumstances) to choose? Will they inform the leadership of the AMBITION study of potential concerns about consent? More generally, how will information flow between the social scientists conducting this qualitative study and the larger clinical trials in which it is embedded? Will the qualitative researchers share their findings in order to change trial procedures? It is unclear what happens if a patient-participant can only do one of the two IDIs, due to losing decisional capacity. Will the data collected still be part of the analysis? Minor comments Page 10, Lines 57-59: "meaning that those who were confused will now no longer be." Now no longer be ineligible? There seems to be a word missing here. Page 12, Line 25: DSL, lead clinician for the trial, is said to be the recruiter of AMBITION trial researchers for this qualitative study. It would be better to have someone independent or at least further down in the trial hierarchy to do this, otherwise it seems hard for researchers to refuse. Page 18: the AMBITION study schema is confusing as it adds reference to an 'economic evaluation study from a societal perspective' which did not figure in the protocol as described in the text.
--	--

VERSION 1 – AUTHOR RESPONSE

Reviewer 1

It is a study with an excellent description of the methodological and very relevant procedures.

Response: Thank you for this positive feedback.

I find it interesting to insert the objective of the manuscript at the end of the introduction.

Response: Thank you for this comment. We feel that the introduction of the manuscript sets the scene, identifies the current gaps in the literature, and builds the rationale for this study. We believe therefore that the manuscripts flows well from the background to the aims and the objectives of the LEOPARD study before then moving on to discuss the methodology in detail.

The authors justified the importance of the analysis “rumors” and “ethical implications”, but it does not bring these two topics to the objectives related to the study. I suggest adding.

Response: Thank you for this acknowledgement. We are absolutely interested in the potential impact of rumours within this study as well as broader ethical implications of conducting research for life-threatening illnesses in a resource limited setting. We believe that our approach to data collection will provide the opportunity for us to elicit rich data on these areas. Our objectives are currently rather broad. Whilst we believe that rumours, for example, may emerge as a theme we do not believe that they warrant their own specific objective as we believe that the broad objectives listed would

encompass this. We therefore politely request to keep the objectives as they are currently stated.

Are detailed interview scripts available? Can they be made available as an attachment to this article? Is there a script for direct observations? Can it also be made available as an attachment to this article?

Response: Thank you for this important point. We do have schedules for the semi-structured interviews (participants, next-of-kin and researchers) and have included these as appendices. These are not exhaustive scripts and are loosely structured in keeping with our general approach to data collection. There is no script for the direct observations which is concerned more with observing the often unpredictable day-to-day activities of the trial.

It is interesting to contextualize the three sites that will be included in the survey.

Response: Thank you. We look forward to comparing and contrasting the data collected from these three very different settings.

Reviewer 2

The qualitative study described is likely to produce valuable information about clinical trial participation among those involved in studies of life-threatening conditions in low- and middle-income settings, such as their trial sites in sub-Saharan Africa. I only have a few comments and questions.

Response: Thank you for the positive summary of the study.

How will decisional (in)capacity, the key inclusion/exclusion criterion for patient-participants, be determined?

Response: This is an important point and we have clarified this further on page 4 and page 9 of the tracked changes version of the manuscript. Specifically, we now state that:

'The decision making capacity of potential participants is assessed by the clinical team who determine if the individual is able to understand the information around the trial, retain that information, weigh up the information to make a decision and communicate that decision. Patients consent for themselves if deemed to have decision making capacity and if they do not, for example if they are confused or comatose, then a surrogate will do so on their behalf.'

And:

'Individuals who upon entry into the AMBITION trial were deemed to have decision making capacity (i.e. orientated) and those who were not (i.e. disorientated), and therefore underwent surrogate consent, will be approached. All participants in the LEOPARD study will need to have decision making capacity to contribute to the IDI, meaning that those who lacked decision making capacity at baseline will have clinically improved and regained that capacity.'

For reasons that the protocol itself states, what will the qualitative researchers do if they find that the quality of informed consent among participants or next-of-kin for the AMBITION study is poor, either in terms of what participants understand or in terms of their ability (given their circumstances) to choose? Will they inform the leadership of the AMBITION study of potential concerns about consent?

More generally, how will information flow between the social scientists conducting this qualitative study and the larger clinical trials in which it is embedded? Will the qualitative researchers share their

findings in order to change trial procedures?

Response: Thank you for raising these two important points which we will address together. We would like to reassure you that this has been considered and is part of the full protocol and the implementation of this study. We would like to apologise for not explicitly stating this in the manuscript and we have done so now. We have added the following to page 11:

'It is anticipated that this study may identify aspects of the AMBITION trial that need to be improved. In order to ensure this a formal reporting process will be followed. Each of the individual social science research assistants will report back to DSL. Any urgent issues that relate to trial conduct and Good Clinical Practice will be communicated through the use of direct communication and reflective summaries written on the day of data collection. In addition, weekly meetings will take place between the social scientists and DSL to discuss less urgent issues. These findings will be communicated either urgently to the Trial Management Group or at their weekly meetings, whichever is deemed appropriate. Additional advice may be sought from JS who is independent of the AMBITION trial. Following this process the team will determine a course of action which may result, for example, in additional training of trial staff or modification of study procedures. This process is of vital importance to ensure that the findings of this study can improve the conduct of the ongoing AMBITION trial. The confidentiality of the participant will be maintained throughout this process so as not to undermine trust in the study.'

It is unclear what happens if a patient-participant can only do one of the two IDIs, due to losing decisional capacity. Will the data collected still be part of the analysis?

Response: Thank you for raising this point. There are a myriad of reasons why one individual may not be able to attend for both interviews. We have added a sentence on Page 9 to address this comment: 'If individuals can only contribute to one IDI, for example due to worsening health or unavailability, then the data from the first IDI will be retained and analysed.'

Page 10, Lines 57-59: "meaning that those who were confused will now no longer be." Now no longer be ineligible? There seems to be a word missing here.

Response: We believe we have addressed this comment in our response above. We now state: 'meaning that those who lacked decision making capacity at baseline will have clinically improved and regained that capacity'.

Page 12, Line 25: DSL, lead clinician for the trial, is said to be the recruiter of AMBITION trial researchers for this qualitative study. It would be better to have someone independent or at least further down in the trial hierarchy to do this, otherwise it seems hard for researchers to refuse.

Response: We acknowledge this important point and thank you for raising it. DSL is a PhD candidate and the LEOPARD study will form his PhD thesis. We have stated explicitly his role within AMBITION and LEOPARD in this manuscript and fully accept the impact of his position on the study. The full study protocol discusses this in detail and we have added some of this text to the edited manuscript on page 10.

'The researcher participant will be assured that they are free to decline participation and are not being interviewed or observed for the purposes of any appraisal or formal evaluation of their role within the team. The purpose of the researcher interviews and observations is to understand the research process and not to criticise individuals. A reflective approach to the research process will be adopted to iteratively refine the data collection methods but also the communication skills of the social scientists.'

Page 18: the AMBITION study schema is confusing as it adds reference to an 'economic evaluation study from a societal perspective' which did not figure in the protocol as described in the text.

Response: Apologies for this confusion. We have modified the figure.

Formatting Amendments

'Strengths and limitations of this study' should consist of 3-5 bullet points. However, more than 5 points were provided. Kindly modify the provided 'Strengths and limitations of this study' to conform with the requirement.

Response: Apologies for this oversight. We have removed what was the third bullet point: 'This study can prospectively provide feedback to improve an ongoing clinical trial', as we feel that the sentiment of this was actually already captured in the second bullet point. We are now left with 5 bullet points.

Many thanks again to the Editorial Team and reviewers for your comments.

VERSION 2 – REVIEW

REVIEWER	Magno, L Universidade do Estado da Bahia, Salvador, Brasil, Departamento de Ciências da Vida
REVIEW RETURNED	21-Jan-2021

GENERAL COMMENTS	The authors' responses were satisfactory.
---

REVIEWER	Rennie, Stuart UNC School of Medicine Charlotte Campus
REVIEW RETURNED	18-Jan-2021

GENERAL COMMENTS	Thank you for responding adequately and in detail to the comments I provided for the previous manuscript version.
---